# IgG Fc Affinity Ligands and Their Applications in Antibody-Involved Drug Delivery: A Brief Review

**DOI:** 10.3390/pharmaceutics15010187

**Published:** 2023-01-05

**Authors:** Chang Yang, Bing He, Hua Zhang, Xueqing Wang, Qiang Zhang, Wenbing Dai

**Affiliations:** Beijing Key Laboratory of Molecular Pharmaceutics and New Drug Delivery Systems, State Key Laboratory of Natural and Biomimetic Drugs, School of Pharmaceutical Sciences, Peking University, Beijing 100191, China

**Keywords:** IgG affinity ligands, antibody delivery, antibody–drug conjugates, drug delivery, active targeting

## Abstract

Antibodies are not only an important class of biotherapeutic drugs, but also are targeting moieties for achieving active targeting drug delivery. Meanwhile, the rapidly increasing application of antibodies and Fc-fusion proteins has inspired the emerging development of downstream processing technologies. Thus, IgG Fc affinity ligands have come into being and have been widely exploited in antibody purification strategies. Given the high binding affinity and specificity to IgGs, binding stability in physiological medium conditions, and favorable toxicity and immunogenicity profiles, Fc affinity ligands are gradually applied to antibody delivery, non-covalent antibody–drug conjugates or antibody-mediated active-targeted drug delivery systems. In this review, we will briefly introduce IgG affinity ligands that are widely used at present and summarize their diverse applications in the field of antibody-involved drug delivery. The challenges and outlook of these systems are also discussed.

## 1. Introduction

Antibodies (Abs), also known as an immunoglobulin (Ig), represent one of the most important classes of biotherapeutic drugs [1]. They are used to treat many diseases, including cancer, allergy, and autoimmune diseases [2]. Antibodies are macromolecules, having a molecular weight of approximately 150 kDa. An antibody molecule consists of antigen-binding fragment (Fab) and crystallizable fragment (Fc) to form a “Y” structure [3]. The Fab region is a mixed light-heavy chain dimer of variable light (V_L_), constant light (C_L_) and variable heavy (V_H_), constant heavy 1 (C_H_1) segments, and the Fc region is a heavy chain dimer composed of constant heavy 2 (C_H_2) and constant heavy 3 (C_H_3) segments (Figure 1). The Fab region determines the specificity and affinity of an antibody to its target antigen; the Fc region can actively complement or bind to the Fc receptors expressed on the surfaces of the immune effector cells, so as to exert the effector functions of the antibodies such as the antibody-dependent cellular cytotoxicity (ADCC) and antibody-dependent cellular phagocytosis (ADCP) [4]. Based on the current understanding of the mechanisms of several therapeutic antibodies, many now believe that Fc-mediated functions will improve the clinical effects of therapeutic antibodies. Therefore, many kinds of antibodies and immunoglobulin-derived products have been developed, such as Fc-fusion proteins. Increasing the products has led to improvements in the upstream processing, which in turn has increased the urgent requirements of the advanced downstream processing. After expression in recombinant mammalian cell cultures, antibodies undergo different purification steps including centrifugation, filtration, and affinity chromatography [5]. Because Fc regions are relatively conserved molecular structures of antibodies, most of the affinity chromatography, which is now widely used in antibody purification processes, is bind to Fc regions for reasons of universality.

Traditionally, IgG Fc affinity proteins, which could non-covalently bind with Fc regions, have been widely explored in antibody purification processes [6]. Protein A (pA) and protein G (pG) have been used due to their high affinity toward different classes of IgG [7]. However, major problems encountered when using bacterial proteins include the high cost, the contamination of the purified antibody samples with traces of these proteins (better known as ligand leakage), and the activity and immunogenicity of antibodies loss caused by harsh elution conditions. Considering the drawbacks, recent decades have witnessed a great effort in the development of new IgG affinity ligands for antibody purification [8,9,10,11]: for example, the protein A/G obtained by fusing protein A with a gene in the Fc-region binding domain in the protein G molecule.

With the wide development of affinity ligands, it is gradually realized that they are not only limited to the purification of antibodies, but also gradually applied to more and more fields, such as researches including recombinant proteins, biomedical engineering, and electronic engineering. For example, there are studies on the use of affinity peptides to immobilize antibodies in the appropriate orientation to the sensor and the use of affinity proteins to increase the half-life of fusion proteins [12,13]. It is worth noting that IgG Fc affinity ligands have received increasing attention in the field of antibody-involved drug delivery due to their biocompatibility, high-affinity selectivity for antibodies or Fc-fusion proteins [14], and stability of non-covalently affinity complexes in physiologically relevant medium [6].

In this review, we will briefly introduce IgG affinity ligands that are widely used at present and summarize the advanced applications of affinity ligands in the field of antibody delivery, non-covalent antibody–drug conjugates (ADC), and antibody-mediated active targeting drug delivery systems. We will highlight some representative work and present the challenge and outlook of the future research and development of antibody-involved drug delivery.

## 2. IgG Fc Affinity Ligands

IgG Fc affinity ligands have already been reviewed in the earlier literature [15]. Therefore, this article will not make a comprehensive review about them, but only representative ligands that are widely used in the field of antibody-involved drug delivery.

IgG affinity proteins were initially developed for application in antibody purification, and the most widely used were the natural affinity proteins of bacteria, such as pA and pG [16]. However, these bacterial proteins have a number of drawbacks, such as high production cost, poor stability, poor affinity specificity, and harsh elution conditions. To overcome these drawbacks, different kinds of immunoglobulin affinity ligands with improved properties have been developed [8], including engineered variants of natural proteins, short peptides, and synthetic low-molecular-weight ligands.

As shown in Table 1, most of the affinity ligands are proteins and peptides, and the binding site is mainly the Fc region of antibodies. It is worth noting that the shortest affinity peptide contains only four amino acids, which means that its immunogenicity does not need to be especially considered when applied in vitro and in vivo. In addition, low-molecular-weight ligands with simple structures, such as 4-mercaptoethylpyridine (MEP), are included. The various affinity ligands exhibit a wide range of affinity abilities, with Kd ranging from 1 × 10^−9^ M to 1 × 10^−4^ M, which means that different ligands can be selected according to different application scenarios. High selectivity and stability, low toxicity and cost, and selectable affinity capacity provide prerequisites for affinity ligands to be used in the field of drug delivery.

## 3. Applications of Affinity Ligands in Antibody Delivery

Monoclonal antibodies (mAbs) are now applied as targeted therapies for cancer, autoimmune and infectious diseases, as well as a range of new diseases. However, systemic administration of mAbs carries the risk of immune reactions, such as anaphylaxis and the production of anti-drug antibodies [37]. In addition, there are many adverse effects associated with mAbs based on their specific targets, including infections and organ-specific adverse events such as cardiotoxicity [38]. Moreover, it has been reported that intracellular proteins have great potential as tumor targets. However, the large size and high surface charge of antibodies severely hinder their internalization into cells [39]. There have been a lot of studies on the use of drug delivery technologies to solve these problems. Most studies have been conducted using hydrogels for local delivery of antibodies to achieve controlled release and mitigate side effects, or using various nanocarrier delivery systems for intracellular delivery of antibodies [40,41].

Compared with systemic administration, local administration at the site of disease allows delivery of higher “effective” doses while enhancing the stability of therapeutic drugs and minimizing side effects. Hydrogel systems, including peptide hydrogels, have proven to be highly biocompatible materials for disease-initiated in situ assembly, programmed degradation, and drug release. However, there are still limitations in current hydrogel-based delivery systems, such as the burst release of drugs [42]. Peptide hydrogels are widely used to deliver antibodies in a sustained and controlled manner because of their simple synthesis, good biocompatibility, low immunogenicity, and low toxicity [43]. By increasing the crosslinking density, the burst release of antibodies out of the network can be limited, but the rate of the antibody release cannot be controlled [44]. Inspired by polymeric gels which have controlled release systems modified with biotin, much research has engineered Fc-affinity ligands into peptide gels to formulate IgG for localized delivery [45].

The EAK16-II, consisting of the sequence AEAEAKAKAEAEAKAK, which self-assembles into crosslinking fibrils in ionic solutions, is often used for local delivery and release of antibodies. However, IgG antibodies are rapidly released. As shown in Table 2, there are many studies based on EAK that are coupled with affinity ligands, thereby achieving controlled release of antibodies. Liu et al. [46] designed an injectable gel of pG_EAK/EAK based on a multivalent display of Fc affinity sites for antibody capture and controlled release (Figure 2a). The functional sequence consists of a truncated pG fused to a duplicated gene of the amphiphilic sequence AEAEAKAK (“EAK”). In vitro experiments demonstrated that the gels could capture IgG without altering the binding affinities and kinetics of pG for IgG. The results of in vivo experiments showed that the retention time of IgG antibodies formulated with pG_EAK/EAK gel was prolonged after injection into both the subcutaneous spaces and footpads of mice. As a demonstration of potential bioengineering applications, thymic epithelial cells (TECs) were mixed with pG_EAK/EAK gel, formulated with TEC-specific anti-EpCAM antibodies. The injected TECs aggregated into functional thymic units in vivo, supporting the development of CD^4+^ and CD^8+^ T cells and Foxp3+ regulatory T cells in mice. The pG_EAK/EAK gels can be used to retain IgG locally in vivo and can be tailored as a scaffold to control molecular and/or cellular therapy deposition. This localized delivery system can reduce immune-related adverse events for many of the immunomodulatory agents currently approved for use in humans. Li et al. [47] developed self-assembling immuno-amphiphiles (IAs) by the direct conjugation of Z33, the protein A-mimicking peptide, to linear hydrocarbons (Figure 2b). The results showed that the immunofibers can bind to human IgG1 with a nanomolar Kd value and have some application potential in the pharmaceutical industry.

Compared to gels that achieve controlled release of antibodies through biotin, heparin, and divalent metal [48,49], gels with coupled antibody affinity molecules do not require redundant modification of antibodies and are suitable for a wide range of IgG antibodies. These affinity gels are easily fabricated in aqueous solution and greatly facilitate in situ encapsulation of bioactive proteins.

**Table 2 pharmaceutics-15-00187-t002:** Antibody Delivery by Different Peptide Sequences.

Application	Sequence	Affinity Ligand
Local Delivery of Antibodies	pG_EAK/EAK [46]	Protein G
	C12-Z33 [47]	Z33
	EAK16-II-EAKH6 [50]	Protein A/G
	EAK-EAKH6 [51]	Protein A/G
	EAK16-II [20]	Z15
Intracellular Delivery of Antibodies	CPD-prtA [52]	Protein A
	CPD-TRIM21 [52]	TRIM21
	CPP-pAd [53]	Protein Ad
	TAT-B2C [54]	B domain
	FcBP-Tat [55]	FCⅢ

The development of antibodies against intracellular drug targets is an effective way to treat diseases. However, the large size and high surface charge of antibodies limit the development and application of intracellular targeting antibodies [39]. To deliver antibodies into living cells, microinjection and electroporation methods have been shown to serve this purpose [56]. However, these methods often cause severe damage to cells [57] or inactivate antibodies during encapsulation or chemical treatment [58], which make them difficult to be applied in vivo [59].

As shown in Table 2, there are strategies to use IgG affinity proteins fused to the genes of cell-penetrating polymers (CPPs) that, upon binding to the Fc regions of IgG, produce non-covalent complexes that can readily cross the cell membrane. Chong et al. [60] fused the Z domain from pA with specific nanomolar binding affinity to the Fc regions and multimeric LK sequence with strong cell-penetrating activity together to make multimeric LK-Z domain carriers (Figure 3a). Moreover, a cathepsin B sensitive sequence was added between the Z domain and CPP to ensure that the delivered IgG would be released from the LK sequence for its function after internalization. The results showed that the LK-Z domain could successfully deliver the IgG into cells at concentrations less than 10 nM and improve the inhibition of the targeted signaling pathways inside live cells. Such individual modifications can increase the cellular uptake of functional antibodies, but it is not clear whether a single antibody in the complex is sufficient to exert the antibody effect. Lim et al. [61] formed a fusion protein from a self-assembling alpha helical peptide with protein A domain B (SPAB), which was able to self-assemble into a Hex nanocarrier (Figure 3b). The positioning of multiple Fc affinity domains on the hexametric core allowed the Hex nanocarrier to tightly bind the antibody with sub-nanomolar affinity. The Hex nanocarriers were capable of binding multiple antibodies in a single carrier, allowing for increased internalization over soluble antibodies without the need to utilize cell-penetrating peptides, with the advantage of low cytotoxicity. The Hex nanocarriers offered many possibilities for future applications, such as the addition of peptide motifs for endosomal escape or the addition of extracellular targeting ligands to facilitate the Hex nanocarrier concentration at the target cells or tissues. 

Du et al. [52] synthesized different cell-permeant bioadaptors (CpA1, CpA2, and CpT) equipped with IgG-binding proteins (proteins A and TRIM21) and cell-penetrating poly(disulfide)s (CPDs), and then simply mixed them with antibodies to form non-covalent complexes (Figure 3c). This complex can be immediately used for the intracellular delivery of antibodies. A series of examples of successful biological applications include antibody-based live-cell imaging of endogenous protein glutathionylation to detect oxidized cell stress, antibody-based endogenous caspase-3 activation and inhibition of endogenous PTP1B activity, and TRIM21-mediated endogenous protein degradation, indicating that this mix-and-go strategy is practical, efficient, and versatile.

Compared to electroporation and microinjection, non-covalently delivered antibodies avoid disrupting cell membranes or reducing cell viability. Moreover, compared to covalent coupling strategies, this “mix-and-go” method not only avoids the complex step of antibody purification, which causes the structural and functional impairments of antibodies, but also allows delivery of more than one antibody or Fc-fusion protein simultaneously in a facile way [54]. Due to the above advantages, the non-covalent binding strategy has great prospects in applications, such as antibody intracellular delivery and combined therapy based on various antibodies.

## 4. Applications of Affinity Ligands in Non-Covalent Antibody–Drug Conjugates

Over the past ten years, antibody–drug conjugates (ADCs) have made considerable progress in the field of cancer therapy. ADCs utilize the high binding specificity and affinity of antibodies to target antigens and specifically direct potent cytotoxic payloads into cancer cells [62]. The structure of an ADC contains four components: the monoclonal antibody, cytotoxic payload, linker, and conjugation site [63]. The linker used in the construction of ADCs is responsible for the formation of stable conjugates in the systemic circulation and the release of the payload after the ADC internalization at the target site [64]. Currently, there are three methods commonly used to bind linkers to antibodies, namely through chemical coupling strategy, using enzyme catalysis [63], and using antibodies with added cysteine residues or designed to be doped with unnatural amino acids [65]. Structural modifications to antibodies may lead to changes in antibody activity, such as the targeting ability, and also pose a challenge for scaled-up antibody production.

Therefore, an alternative approach to the production of homogeneous ADCs using unmodified antibody scaffolds would be advantageous [66]. As discussed in the previous sections, there are many moieties with high Fc affinity. It is simpler and easier to conjugate drugs to these moieties to form Fc-binding moieties-drug conjugates than to chemically conjugate payloads to antibodies. The drug payloads could be non-covalently conjugated into antibodies by means of the high binding affinity between the Fc affinity ligand and Fc region of the antibody. A significant advantage of this approach is that it can quickly conjugate drugs to site-specific Fc regions to obtain homogeneous non-covalent conjugates. Given the relatively conserved sequence of IgG Fc regions, it is a potentially scalable or broad-spectrum strategy [66].

As shown in Table 3, several studies have used different affinity ligands to generate non-covalent ADCs. Gupta et al. [6] have developed multivalent and affinity-guided antibody empowerment technology (MAGNET) to engineer the non-covalent ADC. Specifically, they attached drug payloads to the IgG antibody via non-covalent binding by using MEP as an affinity ligand. Using gemcitabine and 5(6)-carboxyfluorescein as model payloads, the authors identified three conserved high-affinity binding sites on the antibody scaffold: two on the light chain of the Fab region and one within the heavy chain of the Fc region. These binding sites were validated using two therapeutic antibodies, cetuximab and trastuzumab. After characterizing the structure of the ADCs, the authors validated the in vitro efficacy of trastuzumab derivatives and cetuximab derivatives in SKOV3 and A549 cell lines, respectively, demonstrating internalization and trafficking of the ADCs. The efficacy of ADCs was subsequently tested in a mouse model of human lung cancer. The cetuximab-gemcitabine conjugate inhibited tumor growth twice as much as cetuximab, but both bound tumors to a comparable extent, suggesting that the difference in tumor growth inhibition was a direct result of the drug payload. 

There are also many studies to construct non-covalent ADC systems using the affinity properties of protein A/G and its variants. Maso et al. [67] constructed a system to achieve greater homogeneity and create versatile and adaptable drug delivery systems based on non-covalent linkage between pA or pG and the antibody. Two conjugates were generated by chemical PEGylation of recombinant staphylococcal pA and streptococcal pG, respectively, at the N-terminal and affinity with the antibody (Figure 4a). In vitro flow cytometry analysis showed high selectivity of both conjugates for specific antigen expressing cells. In addition, the non-covalent ADC systems based on trastuzumab and pG, conjugated with a 20 kDa PEG carrying the toxin Tubulysin A, showed a significant cytotoxic effect on the cancer cell lines overexpressing HER2/neu receptors. Using this approach means that the number of drug molecules delivered by a single antibody can be strictly controlled, and as only the Fc region is involved in Fc-binding modules, this ensures that the Fab regions remain free to interact with the target antigen. Muguruma et al. [14] established a novel probe for non-covalent ADCs by synthesizing the Z33-conjugated plinabulin, an antitumor agent (Figure 4b). The synthesized hybrid showed a binding affinity against the Herceptin and the anti-CD71 antibody. The conjugate was able to specifically induce the death of cells with highly expressed antigens, demonstrating that the non-covalent ADC has potential as an antitumor agent.

Rapid generation of homogeneous ADCs by simply mixing antibodies with the payload of the modified affinity sequence avoids the need for antibody modification and destruction. In addition, this strategy can be extended to a wide range of therapeutic molecules as well as diagnostics, with potential uses beyond the treatment of cancer.

## 5. Applications of Affinity Ligands in Active Targeting Drug Delivery Systems

Drug delivery systems (DDSs) are widely used in pharmaceutical research and clinical settings to improve the effectiveness of diagnostic agents and drugs. There are various types of DDSs, including simple macromolecules (such as antibodies, soluble synthetic polymers, and biodegradable polymers) and particulate multicomponent structures (such as nanoparticles, microcapsules, microparticles, cells, cell ghosts, and erythrocytes) [69]. The use of DDSs can overcome the problems of poor aqueous solubility, low bioavailability, and non-specific distribution of conventional drugs in vivo. DDSs can achieve active targeting by attaching targeting ligands, such as monoclonal antibodies, targeting peptides, folate, aptamers, etc., to their surfaces [70]. Antibodies can be attached to the surface of the carrier by covalent coupling, physical and/or hydrophobic adsorption, or affinity of biotin to streptavidin [71]. These methods may require pre-modification of the antibody, increasing process complexity. It is not possible to control the site-orientation of the antibody and where the antibody may contain multiple sites for ligation reactions [72].

Take antibody-modified liposomes, for example, also known as immune-liposomes. They can selectively deliver encapsulated drug “carriers” to cells through the interaction of cell surface proteins with antibodies [73]. However, the preparation of immune-liposomes for each target protein using conventional methods requires chemical modification of antibodies and phospholipids, which is time-consuming. As shown in Table 4, to address these issues, some studies have proposed the use of linkers that can bind non-covalently to the Fc of the antibody for introducing the antibody into the carriers. PAR28 is a pA derivative with a strong affinity for various IgG isoforms and high stability. Hama et al. [74] firstly introduced PAR28 into liposomes by means of PAR28-conjugated phospholipid to prepare PAR28-PEG-liposome (Figure 5a). Then, the liposomes can be further modified by anti-CD147 and anti-CD31 antibodies within 1 h, which can be specifically taken up by CD147 and CD31-positive cells, respectively. The cellular amount of doxorubicin delivered by anti-CD147 antibody-modified liposome was significantly higher than that of liposomes modified by isotype control antibodies. It is worth noting that PAR28-PEG-liposome can be easily and rapidly modified with various antibodies on its surface, which then allows for antibody-dependent selective drug delivery.

Bionanocapsules (BNCs) are hollow nanoparticles composed of protein L (the hepatitis B virus surface antigen) that have a specific affinity for human hepatocytes and are a safe DDS [76]. However, delivery of BNCs is limited to hepatocytes. To give BNCs more targeting possibilities, Tsutsui et al. [77] replaced the pre-S1 peptide with a specific binding affinity with the receptor on human hepatocytes, with the antibody affinity motif of protein A and made the hybrid BNCs linked to anti-human the EGFR antibody recognizing EGFRvIII. The hybrid BNCs were efficiently delivered to glioma cells but not to normal glial cells. The authors confirmed the specific delivery of the hybrid BNCs to brain tumors in an in vivo brain tumor model. In addition, Iijima et al. [75] developed a ∼30 nm bionanocapsule (ZZ-BNC) consisting of a tandem form of hepatitis B virus envelope L protein and protein A-derived IgG Fc-affinity Z structural domain (ZZ-L protein) for tethering antibodies in a directionally immobilization manner (Figure 5b). The authors demonstrated that anti-human epidermal growth factor receptor IgG conjugated ZZ-BNC was endocytosed by human epidermoid carcinoma A431 cells and showed a significant increase in cellular uptake and lysosomal content, cytotoxicity, and tumor suppression compared to α-hEGFR IgG. These results suggest that ZZ-BNC is a promising nano-scaffold to improve the therapeutic efficacy and reduce the dose of antibody drugs.

In another studies, Dostalova et al. [78] used the affinity heptapeptide HWRGWVC to couple antibodies against the prostate-specific membrane antigen (PSMA) to the doxorubicin (DOX)-encapsulated protein apoferritin (APO). Similarly, this strategy, in theory, allows us to modify APO with various antibodies to target a variety of tumors. Ji et al. [79] constructed a model system: anti-Tumor Necrosis Factor (anti-TNFα) is non-covalently bound to the surface of long-lived red blood cells (RBCs) by modifying protein A on the surface of RBCs, while masking the Fc region of the antibody. Conversion of RBCs into therapeutic delivery vehicles would enhance the circulation life of immunoglobulin-based therapeutics while avoiding harmful immune responses.

**Table 4 pharmaceutics-15-00187-t004:** Different active targeting drug delivery carriers and their applications.

Carriers	Linker	Application
Bionanocapsules (BNCs) [75,77,80,81,82]	C2 domain of protein G	As promising drug delivery system nanocarriers for targeting delivery to macrophages.
	Z domain	ZZ-BNCs displaying α-DC or α-hEGFR antibodies on the surface of BNCs and enhancing the therapeutic efficacy to target cell.
Virus-like particle (VLP) of JC polyomavirus (JCPyV) [83]	Z domain	VLP-Z can be armed with cell-specific antibody and be used to deliver therapeutic genes to target cells.
Long-lived red blood cells (RBCs) [79]	Protein A	The RBC-PEG-SpA-antibody complexes can circulate in vivo and efficiently scavenge target antigen molecules.
Gold nanoparticles (AuNPs) [84]	Z domain	AuNP-antibody conjugates (immuno-AuNPs) are of particular value in biomedical applications such as immunoassays for the detection of target antigens.
Adenovirus (Ad) [85,86,87]	C domain of Protein A	The complex can efficiently deliver transgenes to target cells by using the cell entry pathway determined by its ligand component.
	Z33	The gene therapy using Ad-FZ33 with anti-EpCAM or anti-EGFR or anti-CEA antibodies is a potentially effective and safe therapeutic strategy for various cancers.
Liposomes [74]	PAR28	PAR28 can be exploited to easily modify the liposomes with various antibodies and selectively enhance drug delivery into specific cells.
Apoferritins [78]	HWRGWV	Encapsulating DOX in prostate cancer-targeted APO lowers the influence of DOX on off-target organs.

## 6. Challenges and Future Directions

In recent decades, the increase within the variety of therapeutic antibodies has led to requirements for optimized antibody purification processing. More and more kinds of Fc affinity ligands are screened and developed, which promotes the success of antibody affinity chromatography. With the continuous emergence of optimized ligands, their applications are not limited to protein purification. Emerging studies have applied IgG affinity ligands to antibody-involved drug delivery. The choice of affinity ligands in different applications has both similarities and differences. The requirements for the antibody-binding specificity and affinity ability of affinity ligands are consistent in the applications of drug delivery and affinity chromatography for antibody purification. The key differences mainly involve the binding stability of antibody affinity ligand complexes and their controlled dissociation in related conditions. In the context of antibody purification, the complex is expected to remain stable in the stage of antibody capture and dissociate under a specific and mild elution condition, while in the case of drug delivery the affinity ligands should bind stably to antibodies during preparation and in storage. When in use, the complex should be stable until they reach the target site in vivo. It would be better that the complex in the site of the disease could dissociate in a controlled manner. Moreover, in order to explore the full targeting capability in drug delivery or disease treatment, it is crucial that the Fab region of the antibody should not disturbed. 

Despite tremendous progress in antibody-involved drug delivery, including antibody delivery, drug delivery, and active targeting carriers using non-covalent affinity with antibodies, there are still many issues to be addressed. The stability of the complexes in vivo is less explored, and the current applications mostly remain validated in in vitro experiments. Whether the non-covalent affinity strategy can be stably applied to in vivo circulation needs to be further explored in detail. Secondly, antibody aggregation is also a non-negligible problem in its application in drug delivery. Several mechanisms have been proposed for protein-induced nanoparticle aggregation [88]. Therefore, it is necessary to explore the effect of affinity ligand modification on antibodies; for example, hydrophobic ligands may lead to severe antibody aggregation [89]. The aggregated antibodies may trigger an immune response when they enter the body, bringing about greater side effects. In addition, some studies have reported that the matrix surrounding the tumor interferes with the entry of carriers even though they are based on active targeting [90]. This requires that the particle size of the antibody-modified carriers needs to be looked at in the future. Changes in the distribution of carriers of different particle sizes in vivo after antibody modification also need to be investigated.

In addition to issues such as stability and aggregation, for intracellular delivery of antibody applications, non-covalent affinity strategies still suffer from issues such as low cytoplasmic delivery efficiency. Whether the delivered antibody can reach the effective dose is an issue to be considered for future applications. For non-covalent ADCs, the stability of the preparation process and the availability of a highly productive, scalable, and robust synthesis process is also an issue to be explored from the perspective of future applications and industry.

In most cases, the binding of affinity ligands to IgG masks the Fc region of the antibody and perhaps prevents the ADCC or CDC effect of the antibody. Therefore, in the future, it will be necessary to screen an alternative non-covalent binding strategy to ensure that the antibodies attach in a favorable orientation while maintaining their biological function and avoid non-specific protein adsorption. In addition, the controlled dissociation of ligands and antibodies at the target site is also a potential strategy to address the issue of Fc function loss.

Although the application of affinity ligands in drug delivery is at the initial stage, there are still some achievements that are something to be proud of, such as the half-life of non-covalent ADCs in blood after intravenous injection being equivalent to that of antibody, nearly 7 days. More importantly, with the development of more types of antibody drugs and the continuous improvement of downstream processes, more and more affinity ligands will be developed in the future, which provides more alternative ligands for antibody-involved drug delivery.

## 7. Conclusions

Following with their wider applications and success in antibody purification, Fc affinity ligands have been gradually used in various fields including drug delivery. In this brief review, we have described the emerging of Fc affinity ligands and highlighted their application in antibody-involved drug delivery. By means of the binding affinity of Fc affinity ligands with the antibody Fc region, the retention of antibody drugs in hydrogels could be improved for more prolonged antibody release. In the respect of intracellular antibody delivery, the cell-penetrating peptides or polymers are facilely introduced into antibodies and facilitate their internalization into cells. In addition, Fc affinity ligands have been applied in the modification of antibodies on many kinds of drug carries as well as in the development of antibody–drug conjugates, both of which for active targeting delivery of the potent cytotoxic payload into the antigen-overexpressed cells. Although some progress has been made in these directions, some problems remain, such as the uncertainty about the stability and safety of this affinity system application in vivo. Overall, the Fc affinity ligand-based drug delivery is still in its infancy. Future research needs to explore the stability and controllability of the non-covalent antibody complexes or systems, and also could develop versatile platforms to expand the application field of Fc affinity ligands in drug delivery.

## Figures and Tables

**Figure 1 pharmaceutics-15-00187-f001:**
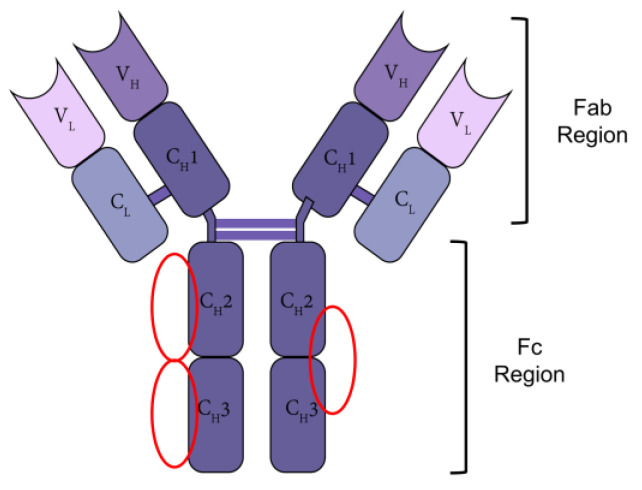
The structure of IgG and the main binding site of IgG Fc affinity ligands; the main binding sites of IgG Fc affinity ligands are shown in red circles.

**Figure 2 pharmaceutics-15-00187-f002:**
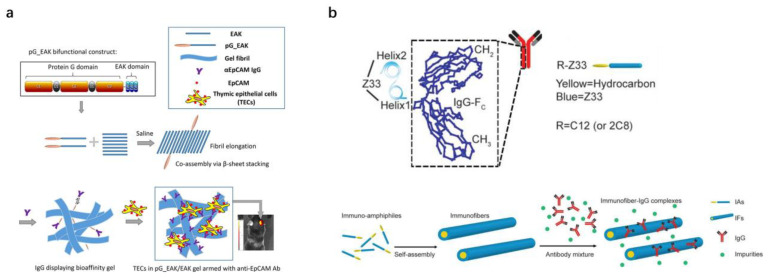
Examples for local delivery of antibodies based on affinity motif. (**a**) Schematic illustration of the design of pG_EAK and the co-assembly of peptide and its bioengineering applications. Reproduced with permission from Ref. [46]. Copyright 2019 *Acta Biomaterialia*. (**b**) Schematic illustration of the Z33 peptide binding to Fc-portion of human IgG1 and the self-assembly of Immuno-amphiphiles and the binding between immunofibers and IgG. Reproduced with permission from Ref. [47]. Copyright 2018 *Biomaterials*.

**Figure 3 pharmaceutics-15-00187-f003:**
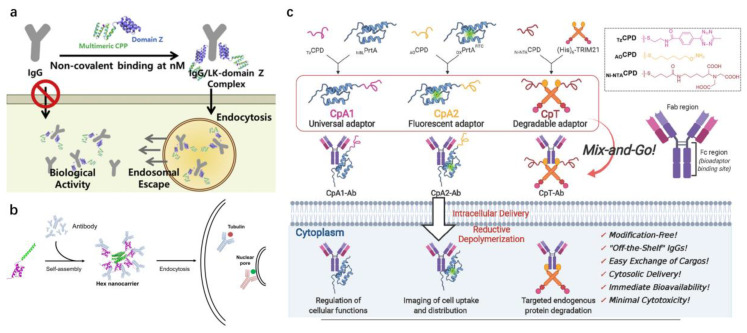
Examples for the intracellular delivery of antibodies based on affinity motif. (**a**) Schematic illustration of the working principle for the intracellular delivery of antibodies at nanomolar concentrations by combination of domain Z and the multimers of LK sequences. Reproduced with permission from Ref. [60]. Copyright 2021 Journal of Controlled Release. (**b**) Schematic illustration of the working principle of the Hex nanocarrier that combines specific protein–protein interactions and nanoparticle benefits. Reproduced with permission from Ref. [61]. Copyright 2017 *Journal of Controlled Release*. (**c**) Schematic illustration of the working principle of the “mix-and-go” approach for CPD-facilitated cytosolic delivery of native functional antibodies with immediate bioavailability. Reproduced with permission from Ref. [52]. Copyright 2020 *ACS Cent Sci*.

**Figure 4 pharmaceutics-15-00187-f004:**
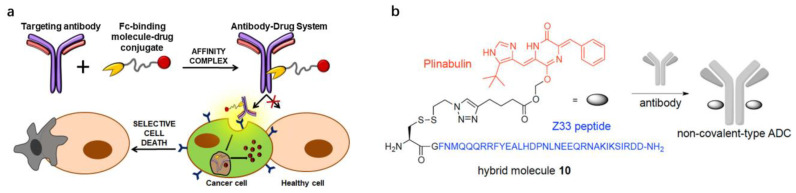
Examples for the construction of non-covalent antibody–drug conjugate based on affinity motif. (**a**) Schematic illustration of the non-covalent ADC systems composed of an anti-tumor agent (Tubulysin A) and IgG affinity protein pG. Reproduced with permission from Ref. [67]. Copyright 2019 Eur J Pharm Biopharm. (**b**) Schematic illustration of the novel probe for non-covalent-type ADC composed of an anti-tumor agent (plinabulin) and IgG affinity peptide Z33. Reproduced with permission from Ref. [14]. Copyright 2016 *Bioconjugate Chem*.

**Figure 5 pharmaceutics-15-00187-f005:**
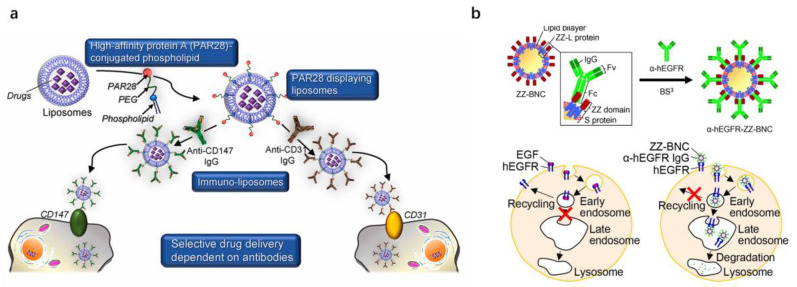
Examples for the construction of active targeting drug delivery carriers based on affinity motif. (**a**) Schematic illustration of the structure of PAR28-PEG-liposomes and construction of immune-liposomes for specific cellular uptake. Reproduced with permission from Ref. [75]. Copyright 2019 Acta Biomaterialia. (**b**) Schematic illustration of the structure of ZZ-BNC and the intracellular intake mechanism of a-hEGFR (left) and a-hEGFR-ZZBNC (right). Reproduced with permission from Ref. [74]. Copyright 2019 *Biochem Biophys Rep*.

**Table 1 pharmaceutics-15-00187-t001:** Commonly used IgG Fc affinity ligands.

Category	Name	Source	Kd	Binding Site
Natural Proteins	Protein A [17]	*Staphylococcus aureus*	7.14 × 10^−7^ M	Fc region (C_H_2–C_H_3 interface)
				Fab region (V_H_ domain)
	Protein G [18]	*Streptococcal groups C and G*	1.49 × 10^−8^ M	Fc region (C_H_2 and C_H_3 interface)
				Fab region (C_H_1)
Engineered Variants of Natural Proteins	Z domain [19]	*B domain of the protein A*	1 × 10^−9^ M	Fc region
	Z38 [20]	*B domain of the protein A*	2 × 10^−8^ M	Fc region (C_H_2 and C_H_3 interface)
	Z33 [19]	*Z domain of the protein A*	4.3 × 10^−9^ M	Fc region
Short Peptides	HWRGWV [21]	synthetic solid-phase random hexamer peptide library	3.8 × 10^−6^ M	C_H_3 domain
	HYFKFD [22]		1.1 × 10^−5^ M	
	HFRRHL [23]		2.6 × 10^−5^ M	
	HWCitGWV [23]		1.08 × 10^−4^ M	
	WQRHGI [24]	peptide search algorithm	3.5 × 10^−7^ M	C_H_2 domain
	FYWHCLDE [25]	biomimetic design strategy	2.4–3.6 × 10^−6^ M	C_H_2–C_H_3 interface
	FYCHWALE [26]		6.1 × 10^−6^ M	
	FYCHTIDE [27]		5.7 × 10^−6^ M	
	NKFRGKYK [28]	spot-synthesized peptide array	1.16 × 10^−7^ M	Fc region
	NARKFYKG [28]		1.54 × 10^−7^ M	
	GSYWYQVWF [29]	phage-display library of random dodecapeptides	0.6 × 10^−6^ M	C_H_2–C_H_3 interface
	DWHW [30]	molecular simulation	1.1 × 10^−5^ M	Fc region
	CEWW [30]		1.2 × 10^−5^ M	
	HEYW [30]		1.7 × 10^−5^ M	
	RRGW [31]	computer design strategy	0.5 × 10^−9^ M	Hydrophobic spots on the surface of lower Fc region
	Fc-III * [32]	phage-display cyclic peptide library	1.85 × 10^−7^ M	C_H_2–C_H_3 interface
	FcBP-2 [33]	peptidomimetics of Fc-III	1.8 × 10^−9^ M	
	Fc-Ⅲ-4C [10]	rational design of Fc-III	2.45 × 10^−9^ M	
Synthetic Low-Molecular-Weight Ligands	ApA [34]	nonpeptidyl mimic for Protein A	1 × 10^−4^ M	Fc region (C_H_2–C_H_3 interface)
	Ligand 22/8 [35]	IgG-binding ligand library	0.714 × 10^−5^ M	Fc region (C_H_2–C_H_3 interface)
	4-mercaptoethyl-pyridine(MEP) [36]	Hydrophobic charge-induction chromatography (HCIC)	1.16 × 10^−7^ M	Fc region and Fab region conserved binding sites

* Fc-III: DCAWHLGELVWCT.

**Table 3 pharmaceutics-15-00187-t003:** Non-covalent antibody–drug conjugates by different linkers and their applications.

Antibody	Linker	Drug	Application
Trastuzumab [14]	Z33	Plinabulin	Tumor-selective non-covalent ADC can be formed in situ and has potential as an antitumor agent.
Trastuzumab [67]	Protein G	Tubulysin A	Obtain more homogeneous non-covalent ADC with significant cytotoxic effect.
Cetuximab [68]	Z33	Dodecaborate	Achieve the intracellular delivery of boron compounds in BNCT.
Cetuximab Trastuzumab [6]	MEP	Gemcitabine	ADCs for the targeted delivery of cytotoxic payloads.

## Data Availability

Not applicable.

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
