# Peer review of "IgG Fc Affinity Ligands and Their Applications in Antibody-Involved Drug Delivery: A Brief Review"

_pharmaceutics, 2023, doi:10.3390/pharmaceutics15010187_

Round 1

Reviewer 1 Report

The authors summarized the current advances and future remarks on IgG Fc affinity ligand in antibody-associated biomedical applications. In general, the paper is well-organized with sufficient contents. I have some minor questions:

1) Table 4. The title is to summarize the nanoparticles functionalized with IgG affinity ligand for targeted drug delivery. However, the contents include red blood cell and Adenovirus. Usually, nanoparticles are defined as synthesized particles with size less than 100 nm, just wondering if the red blood cell carrier refers to a cell or just nanoparticles with cell-components (cell membrane) or not? 

2) Nanoparticle-based delivery system in developing mRNA vaccines has been extensively investigated, this part can be further expanded (e.g., advances on CONVID-19 and cancer vaccines, are there existing system grafted with specific IgG ligands for specific cell targeting?)  

Reviewer 2 Report

The review describes briefly the family of IgG Fc affinity ligands and their applications in drug delivery. The review is very well written and easy to read and understand. The topic is new and important for medicine (first of of for anticancer drug discovery). The article is well illustrated and the data are discussed in details using 90 references. As for English, I am not native speaker, for me English is acceptable. Everything is clear written. I think the paper can be published after minor remarks.

Minor:

It would be interesting for readers if the authors would add the brief chapter (or table) on modern ungoing / recruiting etc. clinical trials based on IgG Fc affinity ligands as drugs (see clinicaltrials.gov).

Author Response

Response to Reviewer 2 Comments

Point 1: It would be interesting for readers if the authors would add the brief chapter (or table) on modern ungoing / recruiting etc. clinical trials based on IgG Fc affinity ligands as drugs (see clinicaltrials.gov).

Response 1: Thank you for your suggestion! As mentioned in the review, although there are various kinds of studies based on IgG Fc affinity ligands, these studies are still in the initial stage and the stability and applications of the complexes in vivo are less explored. Unfortunately, there are no undoing/recruiting clinical trials on clinicaltrials.gov, but we will always pay attention to the research progress. Thank you again for your suggestion.